# High Resolution Distribution Dataset of Double-Season Paddy Rice in China

**Baihong Pan [1], Yi Zheng [1], Ruoque Shen [1] , Tao Ye [2], Wenzhi Zhao [2] , Jie Dong [3], Hanqing Ma [4] and Wenping Yuan [1,*]**

[1] Southern Marine Science and Engineering Guangdong Laboratory, School of Atmospheric Sciences, Sun Yat-sen University, Zhuhai 519082, China; panbh3@mail2.sysu.edu.cn (B.P.); zhengy263@mail2.sysu.edu.cn (Y.Z.); shenrq3@mail2.sysu.edu.cn (R.S.)

[2] Faculty of Geographical Science, Beijing Normal University, Beijing 100875, China; yetao@bnu.edu.cn (T.Y.); wenzhi.zhao@bnu.edu.cn (W.Z.)

[3] College of Geomatics and Municipal Engineering, Zhejiang University of Water Resources and Electric Power, Hangzhou 310018, China; 201531480022@mail.bnu.edu.cn

[4] Northwest Institute of Eco-Environment and Resources, Chinese Academy of Sciences, Lanzhou 519082, China; mahq@llas.ac.cn

[*] Correspondence: yuanwp3@mail.sysu.edu.cn

**Abstract:** Although China is the largest producer of rice, accounting for about 25% of global production, there are no high-resolution maps of paddy rice covering the entire country. Using time-weighted dynamic time warping (TWDTW), this study developed a pixel- and phenology-based method to identify planting areas of double-season paddy rice in China, by comparing temporal variations of synthetic aperture radar (SAR) signals of unknown pixels to those of known double-season paddy rice fields. We conducted a comprehensive evaluation of the method's performance at pixel and regional scales. Based on 145,210 field surveyed samples from 2018 to 2020, the producer's and user's accuracy are 88.49% and 87.02%, respectively. Compared to county-level statistical data from 2016 to 2019, the relative mean absolute errors are 34.11%. This study produced distribution maps of double-season rice at 10 m spatial resolution from 2016 to 2020 over nine provinces in South China, which account for more than 99% of the planting areas of double-season paddy rice of China. The maps are expected to contribute to timely monitoring and evaluating rice growth and yield.

**Keywords:** early rice; late rice; double-season rice; time-weighted dynamic time warping; synthetic aperture radar; planting area; remote sensing

## 1. Introduction

Paddy rice occupies more than 9% of the global cropland area [1] and is a staple food resource for more than half of the world population [2], and, therefore, plays an important role in supporting food security [3–6]. Paddy rice is an important water consumer and greenhouse gas emitter. The global average water consumption of paddy rice is 1325 m$^3$ ton$^{-1}$, and the global water requirement of rice production is estimated to be 784 billion m$^3$ yr$^{-1}$ [7]. In China, irrigating paddy rice needs 1.61 and 2.88 times more water than wheat and maize, respectively [8]. Paddy rice fields are also important sources of greenhouse gasses (i.e., methane, nitrous oxide) due to long-term flooding conditions [9]. Global estimates showed that paddy rice emits about 36 million tons of CH$_4$ and contributes 2.5% ($-0.1$ W·m$^{-2}$) to radiative forcing [10]. Therefore, the accurate identification of paddy rice is quite important for monitoring food security, evaluating water resources, and accounting for greenhouse gas emissions.

Satellite-based methods are widely used to identify planting areas of paddy rice across regional and national scales as they provide spatially and temporally continuous observations [11–18]. Numerous efforts have been made to distinguish paddy rice from other crop types using optical and synthetic aperture radar (SAR) datasets separately or

together [19,20]. Initially, traditional supervised and unsupervised methods were widely used for the classification of paddy rice [21,22]. More recently, machine learning methods such as random forest, support vector machine, and deep learning have been increasingly used for identifying paddy rice [23–28]. Thorp, et al. [29] used deep neural network methods with SAR and optical data from Sentinel 1 and 2 to produce multitemporal maps of paddy rice production stages across West Java, Indonesia. Based on the random forest algorithm and Sentinel data, Fiorillo, et al. [30] mapped lowland rice crop areas in the Sédhiou region (Senegal) from 2017 to 2019. However, the accuracy of these methods strongly depends on the number of training samples [31], which are difficult to obtain and update on a large scale [32].

An alternative method for mapping paddy rice takes advantage of its unique phenological characteristics [33]. The most important feature differentiating paddy rice from other crops is the flooding during the growing season [34]. Previous studies developed an automated mapping algorithm by using satellite-based water and vegetation indexes to detect the phase of flooding and open-canopy of paddy rice [11,12]. This phenology feature has been used for mapping paddy rice in Southern China and Southeast Asia using MODIS data. However, the planting areas of paddy rice are usually distributed in humid regions with frequent cloud cover, which largely limits the availability of optical data [35–37]. MODIS data can provide more cloud-free images because of high temporal resolution, but can't capture the heterogeneity of small and fragmented farmlands. In contrast, high spatial resolution optical remote sensing datasets have low temporal resolution, and are therefore severely impacted by the presence of clouds for mapping paddy rice in humid areas [36,37].

In contrast to optical remote sensing, SAR signals penetrate through clouds and, therefore, can be used under various weather conditions [38–40]. SAR systems are active sensors that emit a radar pulse and record the land surface signal return at the satellite. Water bodies are specular reflectors of the radar pulse, resulting in a minimal or no signal to be returned to the satellite [41,42]. During the flooding and rice transplanting periods, the canopy of rice seedlings is not closed and, therefore, surface waters dominate the satellite response with low values. Guo, et al. [43] added four SAR features to capture the flooding signals, reducing the limitations of lacking optical data during the flooding and transplanting periods. There have been several studies that have used SAR data to map paddy rice, such as in Shanghai of China [44] and Camargue of France [45]. The SAR-based methods were shown to be an effective method for identifying rice areas across different latitudes and planting systems [46,47].

About 25% of global rice production takes place in China. The most common cropping type is double-season rice (early rice and late season rice) and its planting area is concentrated in nine southern provinces. In spite of its importance, there is no high spatial resolution double-season paddy map (i.e., 10 or 30 m) covering the entire region. The objectives of this study are to fill this gap by: (1) developing a pixel- and phenology-based algorithm to classify double-season rice using SAR images; and (2) mapping the distribution map of double-season paddy rice over nine provinces in South China.

## 2. Materials and Methods

### 2.1. Study Area

In China, there are two rice cropping systems: one-season rice and double-season rice. This study aims to identify the planting areas of double-season rice over nine provinces (Table 1; Figure 1), which account for more than 99% of the planting areas of double-season rice in China according to the statistical data from 2016 to 2019. Double-season rice includes two types of cropping patterns: early rice-late rice (Type I) and other crops-late rice (Type II). Type I accounts for 91.45% of areas of double-season rice averaged over all provinces (Table 1).

**Table 1.** Statistical area of early rice and late rice averaged from 2016 to 2019.

| Province | Early Rice (10³ ha) | Late Rice (10³ ha) |
|---|---|---|
| Anhui | 195.03 | 197.15 |
| Fujian | 115.12 | 248.59 |
| Guangdong | 845.05 | 953.09 |
| Guangxi | 799.32 | 839.78 |
| Hainan | 124.03 | 119.92 |
| Hubei | 176.45 | 193.80 |
| Hunan | 1317.08 | 1370.96 |
| Jiangxi | 1219.70 | 1320.21 |
| Zhejiang | 92.22 | 97.33 |
| Sum | 4883.99 | 5340.83 |

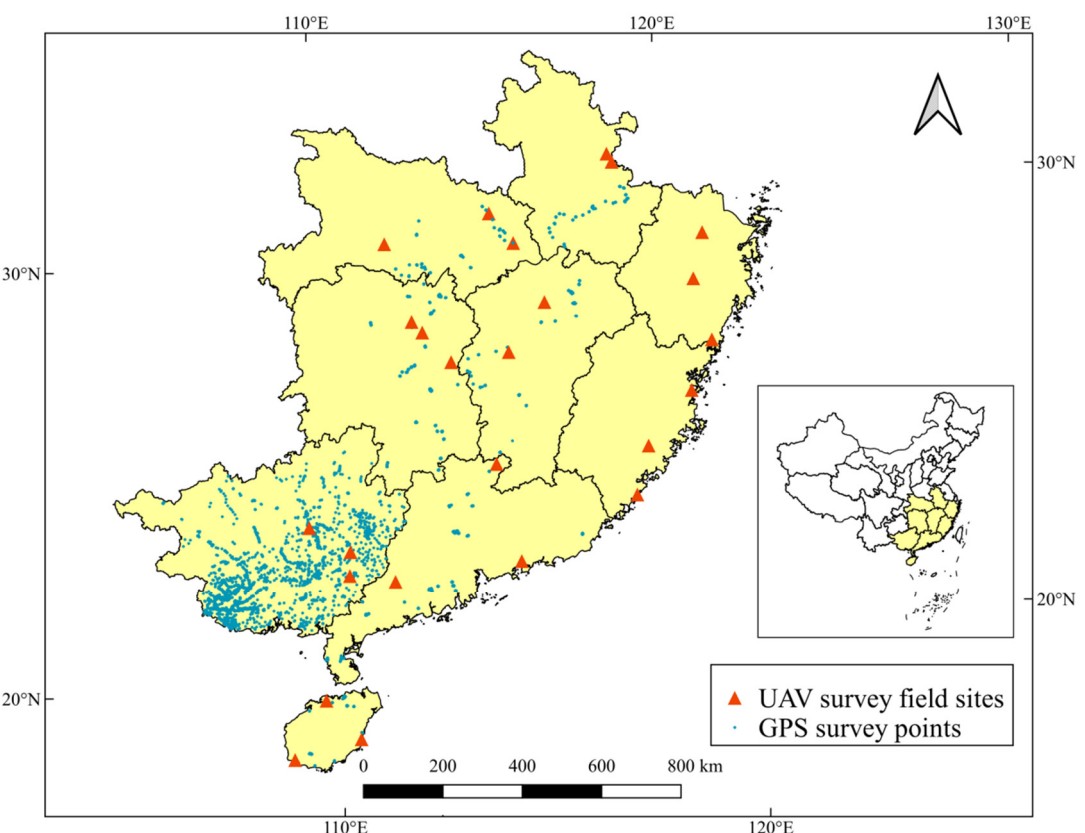

**Figure 1.** Study area spans nine provinces over China (the yellow region). The solid black lines represent the boundary of the provinces. The red triangles indicate 25 unmanned aerial vehicle (UAV) field survey sites in 2018, and each site covers 1 km² field area. The blue dots indicate GPS survey points from 2018 to 2020.

### 2.2. Time-Weighted Dynamic Time Warping Method

In this study, we used the time-weighted dynamic time warping (TWDTW) method to identify the planting area of early rice and late rice. The TWDTW is an upgraded version of the DTW algorithm [48,49]. The DTW algorithm calculates the degree of dissimilarity between two time series (we assume the satellite pixel time series X is the known early rice or late rice field and series Y is that of an unknown land cover pixel) by warping the series Y by adjusting the time dimension and find the minimally modified path to X, which represents the degree of dissimilarity between the two series. Considering the phenological changes of different land covers, Maus, et al. [50] optimized the DTW with a time constraint to develop TWDTW, which avoids excessive warping and thus improves the reliability of the DTW method for crop area recognition [31].

### 2.3. Methods for Identifying Double-Season Paddy Rice Fields

To identify planting areas of early rice and late rice, we need to distinguish them from other land cover types, including non-vegetated land, forest, grassland, other croplands, and water-related cover types. Non-vegetated land includes built-up and barren surfaces. This study used the FROM-GLC10 dataset (see Section 2.6), which classifies non-vegetated land surfaces, forest and grassland with high accuracy [51], to mask them out and exclude them from further processing. To distinguish double-season rice from other crops and water-related cover types, we used the TWDTW method to compare phenological features extracted from seasonal variations of SAR signals.

The definition of rice growth stages is a prerequisite for identifying paddy rice [52]. The growth of rice is commonly divided into four stages [53–55]: (1) The nursery stage (–1 month, from sowing to transplanting); (2) the vegetative stage (1.5–3 months, from transplanting to panicle initiation, including tillering); (3) the reproductive stage (–1 month, from panicle initiation to flowering, including stem elongation, panicle extension and flowering); (4) the ripening stage (–1 month, from flowering to full maturity, including milk stage, dough stage, and mature grain). To highlight the flooding signal and simplify the description of rice growth stages, based on the above four stages, the growth of rice was divided into three stages for this study (Figures 2 and 3): (1) The flooding-transplanting stage (the nursery stage); (2) the growing stage (the vegetative and reproductive stage); (3) the harvest stage (the ripening stage). The rice cropping calendar (Figure 3) was obtained from the field surveys that were conducted from 2018 to 2020 (Section 2.5).

The most important phenological feature of paddy rice is the mixture of surface water and rice seedlings. During the flooding-transplanting stage, the rice canopy is not closed, and surface water contributes substantially to the satellite signals. Water bodies are a specular reflector of the radar pulse of SAR systems, resulting in a minimal or no signal returned to the satellite [41,42]. During the flooding-transplanting stage, the fields of paddy rice are a mix of flooded soils and sparse rice plants, and therefore, the VH (dual-band cross-polarization, vertical transmit/horizontal receive) backscatter coefficient of SAR is quite lower than that of other stages and natural vegetation [56,57]. Subsequently, the VH of rice increases sharply with the growth of rice plants. Therefore, the temporal changes of VH show a "V"-shape curve through the flooding-transplanting and early growing stages (Figure 2).

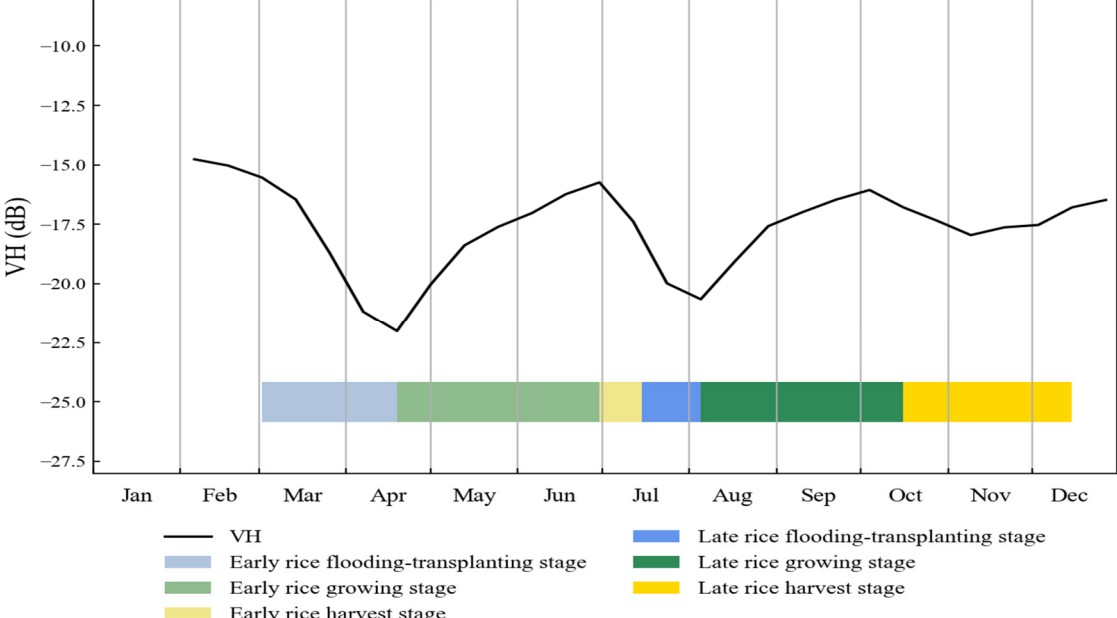

**Figure 2.** The temporal series of satellite-based VH at a double-season paddy rice sample site. VH: Dual-band cross-polarization vertical transmit/horizontal receive backscatter coefficient of SAR.

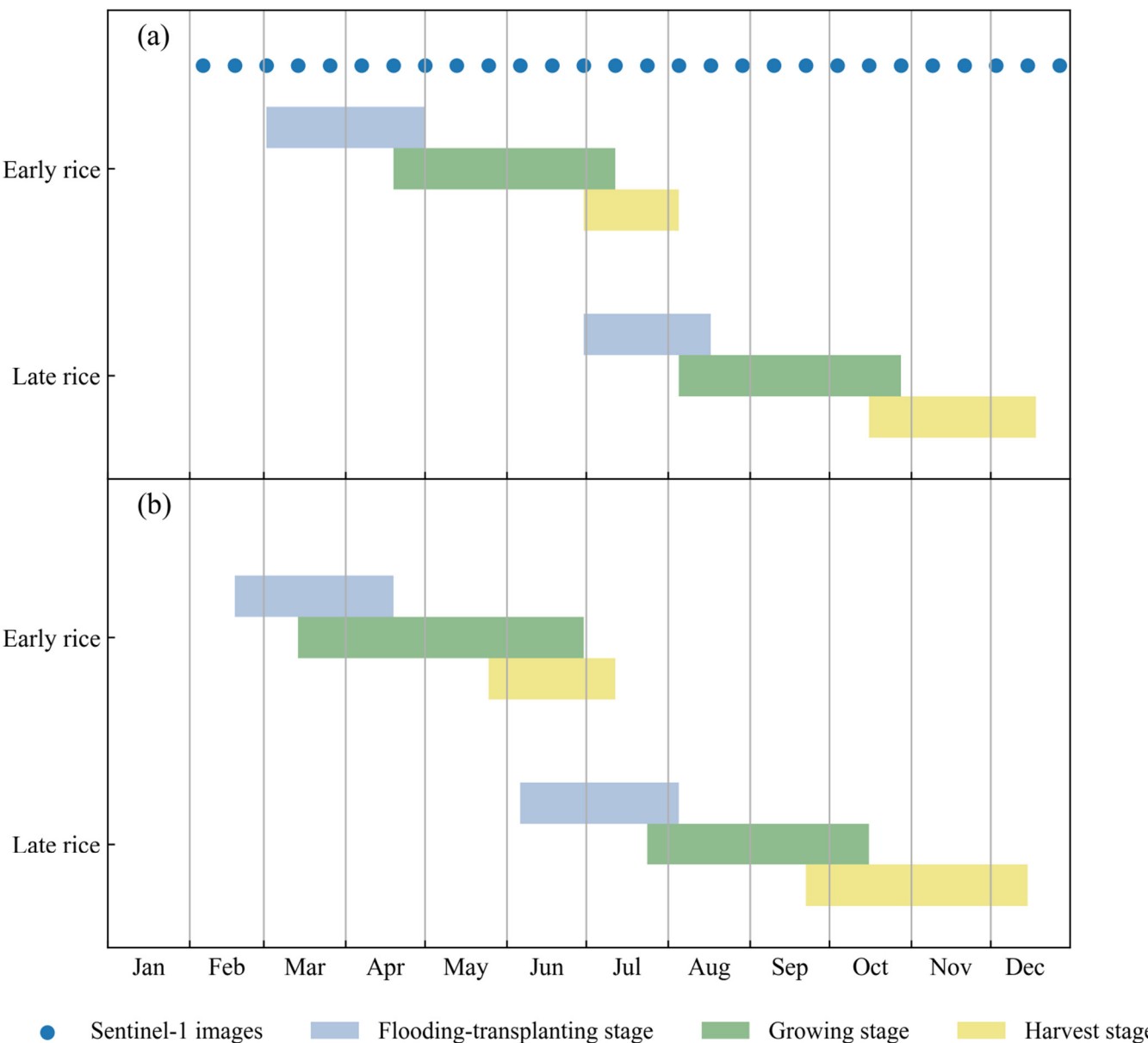

**Figure 3.** Rice cropping calendar and Sentinel-1 image acquisition dates at Guangdong, Guangxi, Hunan, Jiangxi, Fujian provinces, Hubei, Anhui, Zhejiang provinces (**a**) and Hainan province (**b**).

Based on the unique temporal features of VH during the early growth periods (Figure 2), this study used TWDTW to compare the VH curve of a given pixel to that of the standard curve retrieved from known rice fields. Dissimilarity values were calculated to indicate the seasonal change differences of VH between each unknown pixel and known paddy rice pixel. Pixels with lower dissimilarity values have a higher probability of being paddy rice. In this study, we used the statistical area of paddy rice at the province level to determine the thresholds of dissimilarity. Specifically, we selected N pixels with the lowest dissimilarity values as paddy rice in a given province, with the total area of all N pixels equal to the census area of paddy rice in the investigated province. However, the timing of the flooding-transplanting stage of the same kind of paddy rice vary throughout the region, owing to the differences in environments, topography and planting system. Therefore, we shifted the standard VH curve through the potential flooding-transplanting and early growing stages of rice (Figure 3).

We used a simple random sampling approach to select 100 field investigated paddy rice pixels in each province and extracted the time series of the VH curves during the potential flooding-transplanting and early growing stages. Then, we first detected the lowest values of the VH curves from the flooding-transplanting stages to first half periods of the growing stages of early rice and generated the standard VH curves for five time periods: the time of the lowest VH value ($T_{LOW}$), two times prior to $T_{LOW}$ and two times after $T_{LOW}$, totally covering about 60 days (Figure 4a, red curve). In the investigated pixels, we used a moving window covering 60 days to extract the time series of VH from the flooding-transplanting stages to first half periods of the growing stages (Figure 3) and calculated the dissimilarity values by comparing them with the above 100 standard curves. Figure 4 shows how to calculate the dissimilarity values of an unknown pixel compared with one standard curve of early rice in Guangdong Province. Firstly, we generated a standard curve of early rice from a known double-season rice pixel (Figure 4a, red curve). Secondly, we extracted the time series of VH from the unknown pixel with a moving window covering 60 days (Figure 4b–d). In Guangdong Province, the flooding-transplanting stages to the first half periods of the growing stages were from March to April (Figure 3a), so there were three moving windows—18 Feb to 7 Apr (Figure 4b, green curve), 2 Mar to 19 Apr (Figure 4c, green curve), 14 Mar to 1 May (Figure 4d, green curve); we extracted three time series from the same pixel (Figure 4b–d, green curve). Thirdly, based on TWDTW, we calculate the dissimilarity values by comparing the standard curve (Figure 4a, red curve) with each of the three time series (Figure 4b–d, green curve). As a result, we obtained three dissimilarity values. Comparing with 100 standard curves, we calculated a total of 300 dissimilarity values for each unknown pixel, and the lowest dissimilarity value (ERmin) was selected to represent the degree of dissimilarity between the unknown pixels and known early rice pixels.

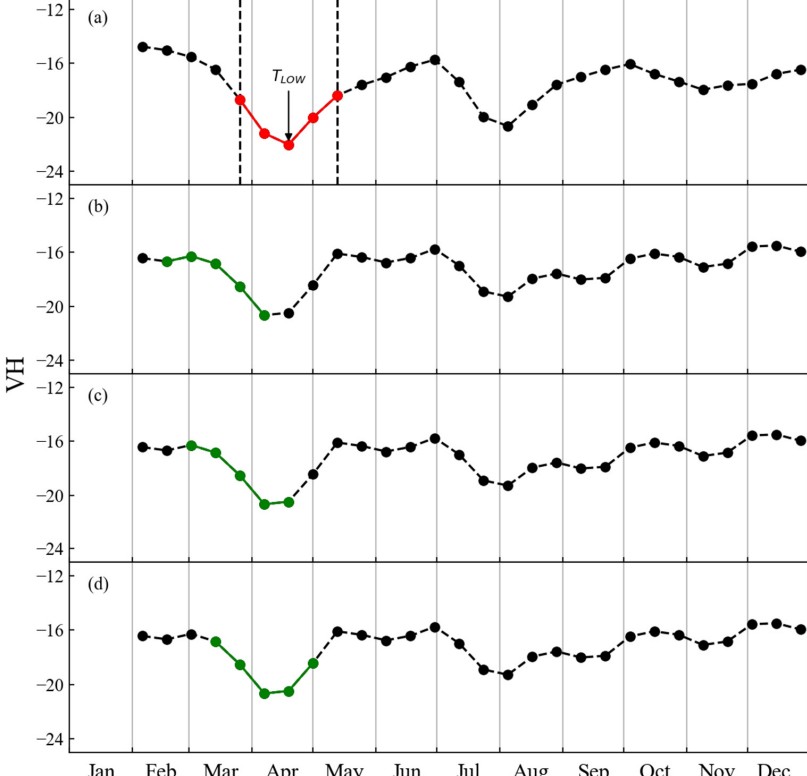

**Figure 4.** The temporal series of VH at a double-season paddy rice pixel (**a**) and red curve means a standard VH curve covering five times (60 days). The temporal series of VH at an unknown pixel applied with three moving windows respectively (**b**–**d**) and green curve means an extracted unknown curve covering five times (60 days).

Following the same process, we also calculated the lowest dissimilarity value of late rice (LRmin) by comparing the standard VH curves of late rice with the VH curve of unknown pixels. The fields planting early rice also plant late rice in the same year, so the study used the sum of ERmin and LRmin to represent the degree of dissimilarity between the unknown pixel and known early rice pixels (ERdist). We first identified planting areas of early rice-late rice type (i.e., Type I) based on the TWDTW method by comparing ERdist of all pixels. Then we chose LRmin to represent the degree of dissimilarity between the unknown pixel and known other crops-late rice pixels (LRdist). In each province, the area differences of late rice and early rice were used to determine the thresholds of LRdist for identifying planting areas of other crops-late rice type (i.e., Type II) based on the TWDTW method.

### 2.4. Satellite Data

This study used the Ground Range Detected (GRD, Level-1) product from Sentinel-1, which has dual-polarized vertical transmission with VV (vertical transmit/vertical receive) and VH (vertical transmit/horizontal receive) bands. We composited the VH data from 2016 to 2020 into corresponding 12-day mean images. Each image was processed to perform thermal noise removal, radiometric calibration, terrain correction and obtain the backscatter coefficient on the Google Earth Engine platform. Even when standard noise-reduction techniques are applied, SAR images contain residual speckle noise due to the interferences between adjacent backscatter returns. To further correct the SAR images for speckle noise, a Savitzky–Golay (SG) filter was applied on the temporal axis for each pixel to smooth the time series. The window size was set to 5, the order and polynomial degree were set to 2 in the SG filter.

### 2.5. Field Data

To obtain the standard seasonal change curve of early rice and late rice, and evaluate the identification accuracy at the pixel level, this study conducted several field surveys during the growing season of early rice and late rice from 2018 to 2020 (Figure 1). First, at 25 investigated sites, we used an unmanned aerial vehicle (UAV) (eBee Extended User Manual, 2015) to take field images in 2018. The UAV was equipped with an RGB camera (Canon S110), which acquired images with a resolution finer than 0.1-m, covering an area of about 1 km$^2$. These images were classified into paddy rice, other crops, natural vegetation and non-vegetation types based on an object-oriented supervised classification as well as field investigations. When using UAV images to generate 10 m surveyed pixels, pixels with 100% double-season rice inside were defined as double-season rice pixels, while 100% non-double-season rice inside were defined as non-double-season rice pixels. In total, these aerial images contained 126,858 pixels at 10 m spatial resolution, of which 48,326 pixels were double-season rice samples, and 78,532 were non-double-season rice samples. Then, we surveyed 18,352 field samples with a hand-held GPS over seven provinces during 2018 and 2020, of which 6127 pixels were double-season rice samples, and 12,225 were non-double-season rice samples (Figure 1).

### 2.6. Land-Cover Dataset and Agricultural Census Data

The Finer Resolution Observation and Monitoring of Global Land Cover (FROM-GLC) product with 10 m resolution was used as a mask to remove non-vegetated land surfaces, forest and grassland [51]. Agricultural census area data of early rice and late rice at the province level during the period of 2016–2019 were acquired from the National Bureau of Statistics of China (2017–2020). The data at the county level during the period of 2016–2019 were acquired from the official website of each municipal-level city statistics bureau.

### 2.7. Accuracy Assessment

The accuracy of identified double-season rice planting areas was evaluated both at pixel and regional scales. First, based on a total of 145,210 ground truth samples retrieved

from field surveys, the study calculated producer's accuracy (PA), user's accuracy (UA) and overall accuracy (OA) to investigate the effectiveness of the method. PA indicates the proportion of ground truth samples properly classified as the target class, and UA indicates the proportion of identified double rice on the classification map that is actually present on the ground. OA is calculated as the ratio of correctly identified samples to total field samples.

In addition, the planting areas of early rice and late rice identified in this study were compared with those obtained from agricultural statistical data at the county level. Three statistical metrics were used:

(1) The coefficient of determination ($R^2$), representing how much variation of statistical area is explained by the identified area.

(2) Root mean square error (RMSE), measuring the deviation between identified and statistical areas, was computed as:

$$\text{RMSE} = \sqrt{\frac{1}{n}\sum_{i=1}^{n}(IA_i - SA_i)^2} \tag{1}$$

where $SA_i$ and $IA_i$ are the statistical area and identified area of the *i*-th county respectively, and n indicates the number of counties in a given province. The unit of RMSE referred to in this study is thousand hectares ($10^3$ ha), and to simplify the description, only the values of RMSE are described below.

(3) The relative mean absolute errors (RMAE), quantifying the difference between identified and statistical areas, was calculated as:

$$\text{RMSE} = \frac{\sum_{i=1}^{n}|SA_i - IA_i|}{\sum_{i=1}^{n}SA_i} \times 100\% \tag{2}$$

## 3. Results

This study generated maps of early rice and late rice over nine provinces in China from 2016 to 2020, and the distribution maps in 2018 are shown as an example (Figure 5). With the updating of the satellite data records, the same method can be used to update the distribution maps of early rice and late rice by the end of each year.

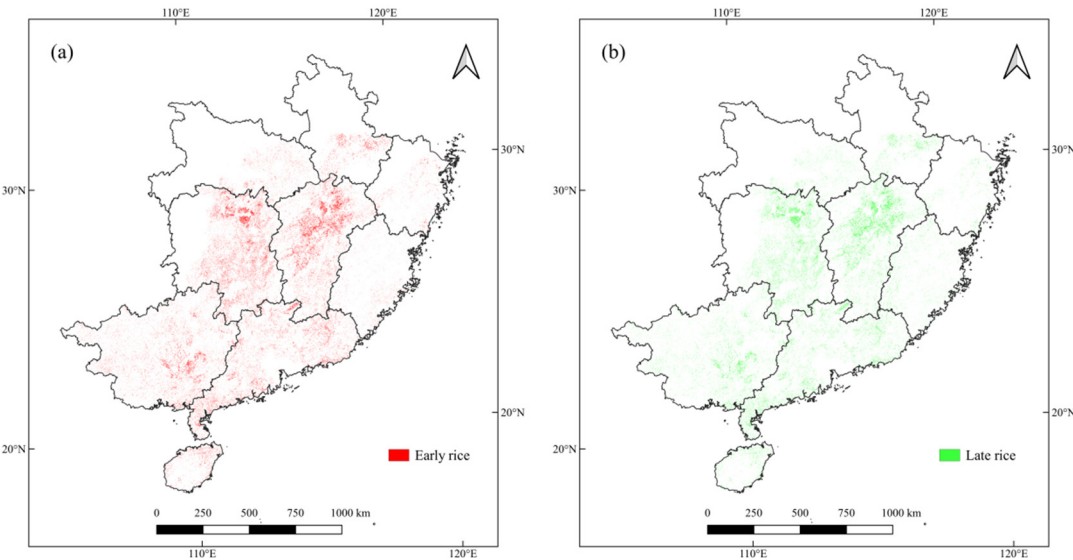

**Figure 5.** Distribution map of early rice (**a**) and late rice (**b**) in 2018.

Our method shows a good performance in identifying the planting areas of early rice and late rice over all the nine provinces. Based on the field survey samples containing

a variety of ground objects, the overall identification accuracy (OA) varied among the nine provinces, ranging from 88.07% to 95.97% for early rice (Table 2), and from 88.25% to 95.68% for late rice (Table 3). The user's accuracy (UA) and producer's accuracy (PA) were high in most provinces. However, for early rice, Hubei Province had the lowest UA of 63.07% and the lowest PA of 73.48% (Table 2). For late rice, Hubei Province achieved the lowest UA of 60.39% and the lowest PA of 74.11% (Table 3).

**Table 2.** Confusion matrix for the identification map of planting areas of early rice in nine provinces from 2018 to 2020.

| Province | Class | Early Rice [1] | Non-Early Rice [1] | User's Accuracy | Producer's Accuracy | Overall Accuracy |
|---|---|---|---|---|---|---|
| Guangdong | Early rice [2] | 11,610 | 1349 | 89.59% | 89.60% | 88.07% |
| | Non-Early rice [2] | 1347 | 8295 | 86.03% | 86.01% | |
| Guangxi | Early rice | 6796 | 1051 | 86.61% | 81.15% | 88.17% |
| | Non-Early rice | 1579 | 12,802 | 89.02% | 92.41% | |
| Hainan | Early rice | 5479 | 639 | 89.56% | 83.75% | 90.44% |
| | Non-Early rice | 1063 | 10,629 | 90.91% | 94.33% | |
| Hunan | Early rice | 12,249 | 2150 | 85.07% | 93.89% | 90.50% |
| | Non-Early rice | 797 | 15,840 | 95.21% | 88.05% | |
| Jiangxi | Early rice | 7219 | 1027 | 87.55% | 97.30% | 91.88% |
| | Non-Early rice | 200 | 6674 | 97.09% | 86.66% | |
| Fujian | Early rice | 771 | 252 | 75.37% | 81.42% | 93.73% |
| | Non-Early rice | 176 | 5629 | 96.97% | 95.72% | |
| Zhejiang | Early rice | 2081 | 277 | 88.25% | 74.83% | 91.75% |
| | Non-Early rice | 700 | 8790 | 92.62% | 96.94% | |
| Hubei | Early rice | 579 | 339 | 63.07% | 73.48% | 95.97% |
| | Non-Early rice | 209 | 12,471 | 98.35% | 97.35% | |
| Anhui | Early rice | 1403 | 101 | 93.28% | 87.80% | 92.85% |
| | Non-Early rice | 195 | 2442 | 92.61% | 96.03% | |

[1] Number of field surveyed samples. [2] Number of identified samples.

**Table 3.** Confusion matrix for the identification map of planting areas of Late rice in nine provinces from 2018 to 2020.

| Province | Class | Late Rice [1] | Non-Late Rice [1] | User's Accuracy | Producer's Accuracy | Overall Accuracy |
|---|---|---|---|---|---|---|
| Guangdong | Late rice [2] | 11,751 | 1450 | 89.02% | 90.69% | 88.25% |
| | Non-Late rice [2] | 1206 | 8194 | 87.17% | 84.96% | |
| Guangxi | Late rice | 6895 | 1099 | 86.25% | 82.33% | 88.40% |
| | Non-Late rice | 1480 | 12,754 | 89.60% | 92.07% | |
| Hainan | Late rice | 5479 | 639 | 89.56% | 83.75% | 90.44% |
| | Non-Late rice | 1063 | 10,629 | 90.91% | 94.33% | |
| Hunan | Late rice | 12,261 | 2206 | 84.75% | 93.98% | 90.36% |
| | Non-Late rice | 785 | 15,784 | 95.26% | 87.74% | |
| Jiangxi | Late rice | 7238 | 1455 | 83.26% | 97.56% | 89.18% |
| | Non-Late rice | 181 | 6246 | 97.18% | 81.11% | |
| Fujian | Late rice | 771 | 252 | 75.37% | 81.42% | 93.73% |
| | Non-Late rice | 176 | 5629 | 96.97% | 95.72% | |
| Zhejiang | Late rice | 2082 | 280 | 88.15% | 74.87% | 91.74% |
| | Non-Late rice | 699 | 8787 | 92.63% | 96.91% | |
| Hubei | Late rice | 584 | 383 | 60.39% | 74.11% | 95.68% |
| | Non-Late rice | 204 | 12,427 | 98.38% | 97.01% | |
| Anhui | Late rice | 1403 | 106 | 92.98% | 87.80% | 92.73% |
| | Non-Late rice | 195 | 2437 | 92.59% | 95.83% | |

[1] Number of field surveyed samples. [2] Number of identified samples.

In addition, this method accurately estimated the areas of early rice and late rice compared to the available agricultural statistical data at the county level for all investigated provinces (Figure 6). The maps reproduce well the spatial variations of the planting area of paddy rice. For early rice in each province (Figure 7), the averaged $R^2$ over three years

of 2016-2019 ranged from 0.50 to 0.87, the averaged RMSE ranged from 1.53 to 6.94, and the average of RMAE ranged from 22% to 59%. For Guangdong, Guangxi, Hunan and Jiangxi Provinces, which occupies 85.61% of the planting area of early rice in China, the classification shows good performance according to three metrics (Figure 7). Fujian and Hubei Province show large classification errors at the county level (Figure 7). For example, the averaged RMAE in Fujian is more than 59%, and $R^2$ is less than 0.50 (Figure 7). Among the four validation years (i.e., 2016–2019), 2016 shows the largest error at the county level (Figure 6).

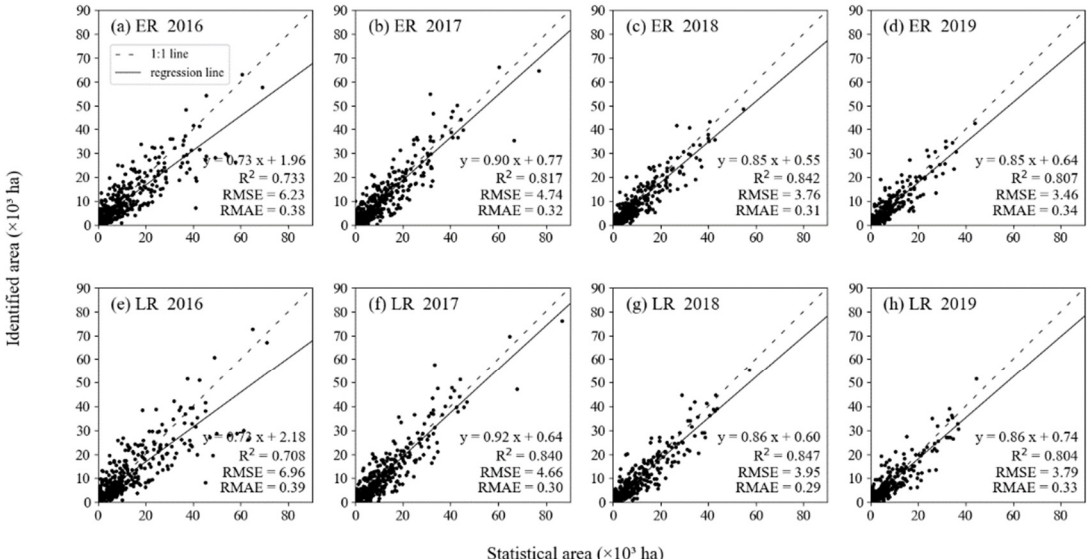

**Figure 6.** County-level comparison of identified and statistical planting areas from 2016–2019, for early rice (**a–d**) and late rice (**e–h**).

The classification accuracy of late rice is similar to that of early rice over almost all provinces. The averaged $R^2$ ranged from 0.56 to 0.86, and the averaged RMAE ranged from 23% to 51% (Figure 8). In four major planting provinces, including Guangdong, Guangxi, Hunan and Jiangxi Provinces, the classification shows high $R^2$ and low identification errors. The largest classification errors are found in Fujian and Hubei Provinces.

To analyze the spatial distribution of isolated patches, we used maps of early rice and late rice for 2018 as an example, and calculated the cumulative area percentage of different patch sizes ranging from 1 pixel to $10^6$ pixels per patch (Figure 9). In nine provinces, early rice patches with only one pixel accounted for 2.31% to 10.59%, but this percentage accounted for 2.31% to 5.97% in the major early rice planting zones including Guangdong, Guangxi, Hunan and Jiangxi Provinces. For late rice, patches with only one pixel accounted for 2.11% to 8.19% in nine provinces but 2.11% to 5.82% in the major late rice planting zones. Considering the cumulative area percentage of early rice patches of 100 pixels, four provinces of the main producing area rank at the bottom, with Guangxi Province with less than 51%; this indicates that the main producing area has mostly large patches, as we would expect for this area. A similar result is found for late rice. It is worth noting that the cumulative area percentage of patches with 100 pixels in Fujian Province drops from 85.29% of early rice to 52.61% of late rice. Because early rice-late rice (Type I) only accounts for 46.31% of the areas of double-season rice in Fujian province (Table 1), this area, and adjacent fields where early rice and other crops are grown during the first cropping season, will plant late rice in the next period. For curves with the cumulative area percentage reaching 100%, four provinces of the main producing area have more large patches, and Jiangxi Province has the largest proportion of large fields.

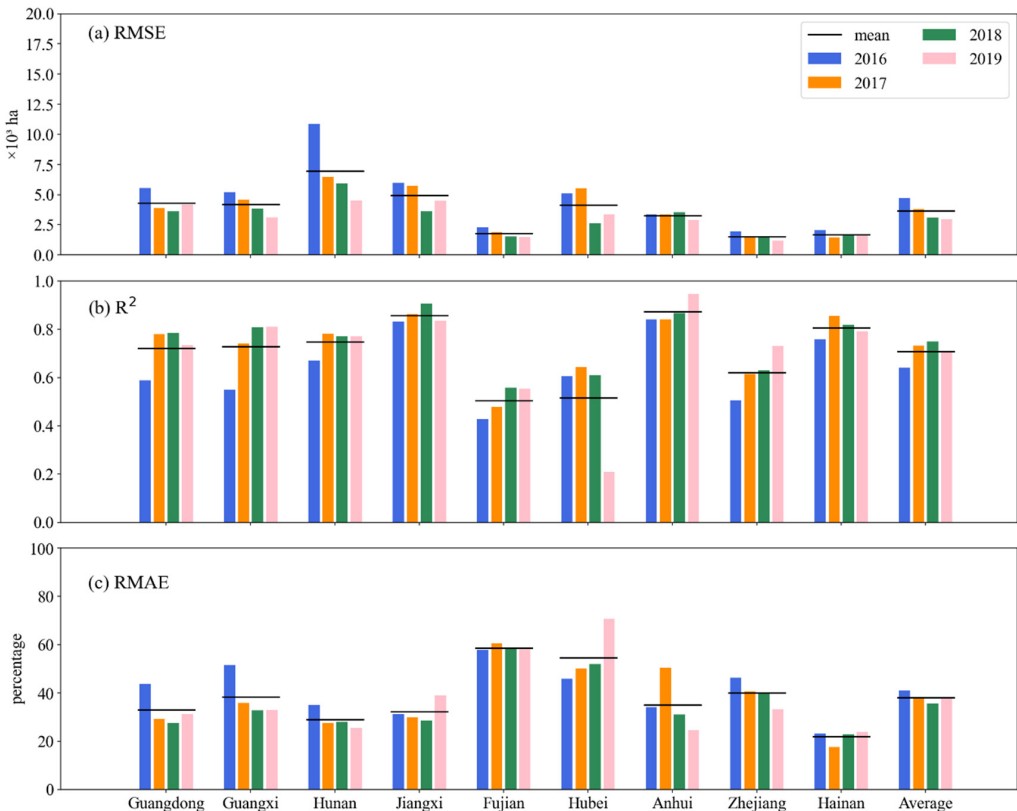

**Figure 7.** The comparison between identified planting areas of early rice and agricultural census areas at county-level for 2016–2019 in all nine investigated provinces.

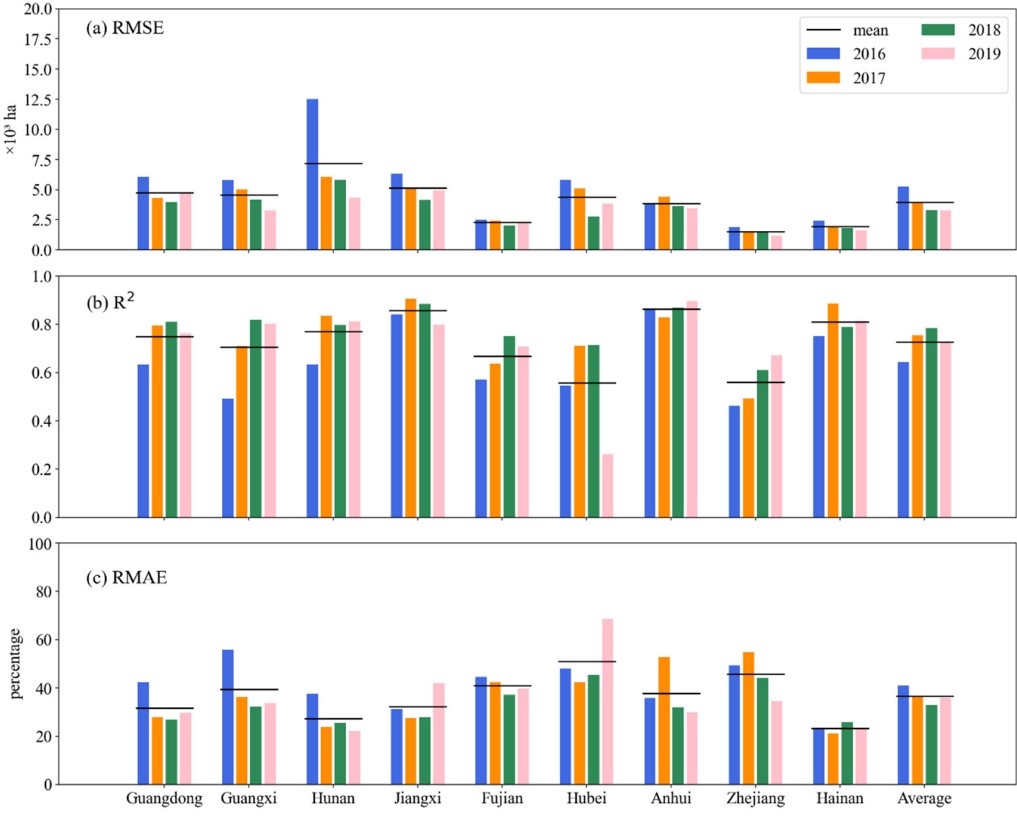

**Figure 8.** The comparison between identified planting areas of late rice and agricultural census areas at county-level for 2016–2019 in all nine investigated provinces.

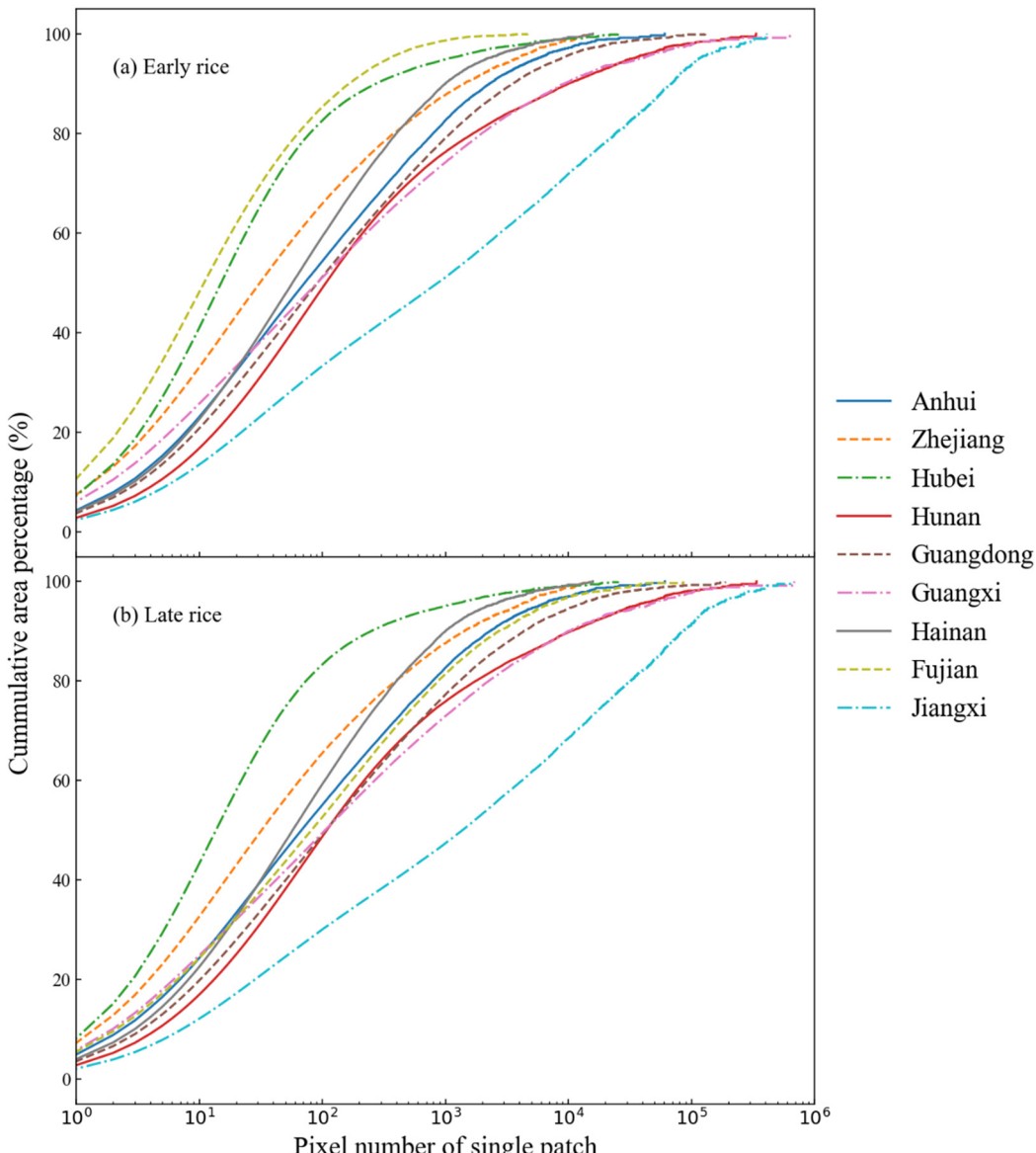

**Figure 9.** Statistics for patches with different pixel numbers in the distribution map of early rice (**a**) and late rice (**b**) in 2018.

## 4. Discussion

Paddy rice is one of the most important crops in the world, and information on its spatial extent is critical for drafting economic and grain subsidy policies. To our knowledge, there are currently no distribution maps for double-season rice of high spatial resolution in China.

In this study, we generated double-season rice distribution maps with a spatial resolution of 10 m for the period of 2016–2020 based on the TWDTW method. Validations based on field surveys and statistical data indicate that the proposed method accurately identifies the early rice and late rice planting areas over all the nine provinces where most of the planting takes place. Unlike machine learning methods that require a large number of training samples [58], our method only requires a small number of field survey samples to produce the distribution maps of early rice and late rice over nine provinces from 2016 to 2020. Our method demonstrates a great skill for mapping 10 m early rice and late rice in South China, where the weather is cloudy and rainy during the key phenological

period, the terrain is mountainous and complex, and fields are fragmented with various planting patterns.

Regional validation (Figures 6, 7c and 8c) shows that the identification accuracy is still low for some of the investigated years and provinces. The identification accuracy is the lowest in 2016 compared to other years and is caused by an inadequate number of Sentinel-1 (Figure 10). Compared to other years, the effective observations in 2016 are only 24 averaged over all nine provinces, which is lower than in other years, when they range from 34 to 40 times. Second, a relatively high RMAE was found in Hubei, Fujian and Guangxi Provinces, where mountainous terrain may be the main cause for the low identification accuracy. For example, in Fujian Province, according to statistical data, 64% of the paddy rice fields are located in the mountainous area of western Fujian province, including Longyan, Sanming and Nanan prefecture-level cities. The SAR backscatter signal is corrupted by the terrain effect, even when complex radiometric terrain corrections are implemented. In addition, mountainous paddy rice fields are small and fragmented, introducing several mixed 10 m pixels without typical VH characteristics [59,60].

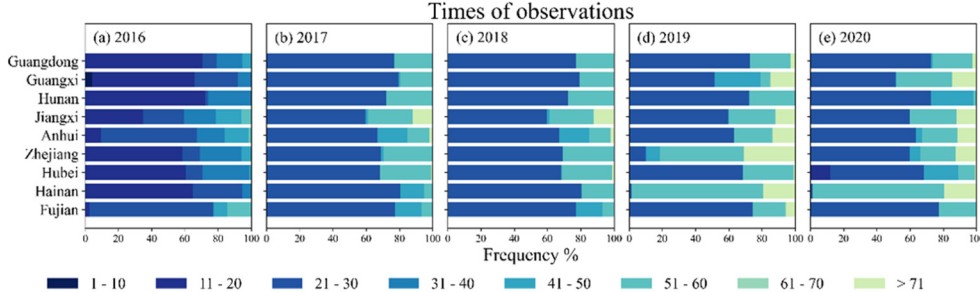

**Figure 10.** Times of Sentinel-1 original observations from February to December during 2016 (**a**) to 2020 (**e**).

The future use of satellite data with higher spatial and temporal resolutions are expected to further improve classification accuracy because higher spatial resolution data will help reduce the number of mixed pixels in mountainous areas and higher temporal resolution data will more easily capture the weak flooding signals resulting from the rapid harvesting and replanting cycles (i.e., the harvest stage of early rice and flooding-transplanting stage of late rice). Future work will also need to use other satellite data to produce rice maps before 2016 when Sentinel-1 data was not available yet. However, data such as those from TerraSAR-X data have a high spatial resolution (3–10 m) and temporal resolution (11-day) but are expensive to use for rice monitoring over large regions. TerraSAR-X data have been used for rice monitoring over smaller regions, like in Spain [61], Mekong Delta, Vietnam [62] and Sanjiang Plain in Heilongjiang Province, Northeast China [63]. Finally, the method can be applied for mapping different planting systems of paddy rice over large regions (i.e., the single-, double- and triple-season paddy rice). The most important difference between the different planting systems is the number of flooding signals in a year. For example, the triple-season paddy rice will have three "V"-shape curves of VH data owing to planting paddy rice three times in a year.

## 5. Conclusions

Based on the available Sentinel-1 images and a time-weighted dynamic time warping (TWDTW) method, this study produced the first 10 m spatial resolution early rice and late rice maps over nine provinces of South China, which account for more than 99% of the planting area of double-season paddy rice in China. Based on 145,210 survey samples, the overall identification accuracy for early rice and late rice were 90.74% and 90.46%, respectively. Compared with the agricultural statistical data at the county level over nine provinces, the maps explain 79.20% and 78.96% of the spatial variability of early and late rice. The timely and accurate 10 m double-season rice maps provide critical information for quantifying methane emissions, water resource management and ensuring food security.

The maps are also useful for commercial companies to make rice production plans or conduct futures transactions while for farmers to formulate personal production plans.

**Author Contributions:** Conceptualization, T.Y. and W.Y.; data curation, B.P., Y.Z., R.S., W.Z., J.D. and H.M.; formal analysis, B.P., Y.Z., R.S., W.Z., J.D. and H.M.; funding acquisition, T.Y. and W.Y.; investigation, B.P. and Y.Z.; methodology, B.P. and W.Y.; project administration, T.Y. and W.Y.; software, Y.Z., W.Z., J.D. and H.M.; supervision, T.Y. and W.Y.; validation, B.P. and R.S.; visualization, R.S.; writing—original draft, B.P.; writing—review and editing, T.Y. and W.Y. All authors have read and agreed to the published version of the manuscript.

**Funding:** This research has been supported by the China National Funds for Distinguished Young Scientists (grant no. 41925001), the National Youth Top-Notch Talent Support Program (grant no. 2015-48), the Changjiang Young Scholars Program of China (grant no. Q2016161), and the Fundamental Research Funds for the Central Universities (grant no. 19lgjc02).

**Data Availability Statement:** The data presented in this study are available in the article.

**Acknowledgments:** The authors would like to thank the reviewers and editors for their constructive comments.

**Conflicts of Interest:** The authors declare no conflict of interest.

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
