# Peer review of "High Resolution Distribution Dataset of Double-Season Paddy Rice in China"

_remotesensing, doi:10.3390/rs13224609_

Round 1

Reviewer 1 Report

The research article " High Resolution Distribution Dataset of Double-Season Paddy Rice in China " brings about a genuine issue to study using time-weighted dynamic time warping (TWDTW), and developed a pixel and phenology based method to identify planting areas of double-season paddy rice in China by comparing temporal variations of synthetic aperture radar (SAR) signals of unknown pixels to those of known double season paddy rice fields. The findings of this article provide distribution maps of double-season rice at 10-m spatial resolution from 2016 to 2020 over nine provinces in South China, which account for more than 99% of the planting areas of double-season paddy rice of China. The research paper is good. However, I would like to suggest the authors can address my comments below for further improvement.

General comments:

Abstract: Abstract is well written and concise. Properly explained the research aims and methods applied to achieve the objectives.

Introduction: The introduction is well written and properly explained the justification of the study. Discussion section may also improve further logically.

Specific comments:

Line 32- Please remove “in addition”. You have not mentioned enough detail before it.

Line 64-66- Please rewrite this sentence again in some meaningful manner.

Line 78- Always write article “The” with abbreviations.

Line 230-233 Please rewrite this sentence again.

Line 238-240- Please improve your sentence structure.

Line 292-294- Please rewrite this sentence again.

Line 343-346- Please short this sentence again. Improve sentence structure.

Line 374-376- Kindly rewrite this sentence again and mention the proper recommendation.

Reviewer 2 Report

The reviewer agrees the authors well revised according to the comments from reviewer.

The manuscript showed the performance of TWDTW method to identify rice planting area. Although the identification needs further improvement, the information might be useful for the readers. 

Following items should be revised: 

  1. Some sentences in M&M and Figure 4 mentioned that the moving window period was for 60 days, but others mentioned 5 consecutive observation of Sentinel-1.  Which is correct? If the observation was missed for 60 days as shown in Figure 10, how did the authors do?
  2. The values of ERdist and LRdist should be shown. 

Author Response

This manuscript is a resubmission of an earlier submission. The following is a list of the peer review reports and author responses from that submission.

Round 1

Reviewer 1 Report

Although the advantage of this method is not well presented in the manuscript, this study may contribute to improve the accuracy of paddy rice mapping. 

Since the authors determined the thresholds of dissimilarity depended on the statistical area of paddy rice, Figure 5 would inevitably obtained. Accordingly, the useful information for the readers is the confusion matrix (Table 2) and errors (Figure 6 and 7) of this method. Following points need to be revised. 

  1. The relatively higher RMAE in Figure 6 and 7 is one of the major concerns. The authors should discuss the reason and the strategy to improve the accuracy.
  2. The description of methods is inadequate. The readers may not reproduce the results. Substantial improvement is required. For example, the reviewer hardly understand "we moved the standard VH (L144)" from where to where; Tlow and 5 VH observations (L154) must be shown in Figure 2 as example; what is "moving windows (L158)"; 
  3. Please describe the definition of rice stages in Figure 2, the way to determine their periods, and how to obtain rice cropping calendar in Figure 3. 
  4. Table 2 and Table 3: which is actual and which is predicted?
  5. Figure 8: Pixel number of single patch must be shown in common logarithm axis to cover larger number of single patch. 
  6.  

Reviewer 2 Report

The study "High Resolution Distribution Dataset of Double-Season Paddy 2 Rice in China" although presented some interesting results and findings. As authors claimed in introduction as still there is no previous study conducted like this is somewhat surprising for me. Although, paper is interesting while i have few queries about the methodology and findings. 

Abstract need to be revised more descriptive to the point (including the research aim, objectives of your project, and the analytical methodologies applied)

The Introduction is well written but I found it weak to describe the research questions with the support of recently published literature.

Line 98. why only "Time-Weighted Dynamic Time Warping Method' used in this study, there may be more other methods/approaches available?

Fig.2. rice crop growth phases should be specific by following any well known scale like BBCH scale where to mention the pehnological stages with respect to number either as authors mentioned in general, early rice flooding...., early rice growing, these are not well defined stages 

Line 149. which method used for randomly selection approach ?

Line 171. Why only "Sentinel-1" data preferred in this study?

Line 175-176. what is need to go for mean images instead of daily?

Line 229. Images from Sentinel-1 are missing while there is only one map for rice distribution. I think, there is need to add more data. 

Fig. 4. are early and late rice crop planted in the same field or different ones, or after one another?

Discussion, Need to more focus on it and especially need to elaborate more in logical discussion and justification. 

what is message for farmers and specially for stakeholders for decision making. 
What about future research work required.
